# Congenital Heart Malformations Masked by Infantile Gangliosidosis—Case Report and Growing Evidence for Metabolic Disease-Associated Aortopathies

**DOI:** 10.3390/diagnostics14050491

**Published:** 2024-02-24

**Authors:** Dana Elena Mîndru, Elena Țarcă, Elena Emanuela Braha, Alexandrina-Ștefania Curpăn, Solange Tamara Roșu, Dana-Teodora Anton-Păduraru, Heidrun Adumitrăchioaiei, Valentin Bernic, Ioana-Alexandra Pădureț, Alina Costina Luca

**Affiliations:** 1Department of Mother and Child Medicine, University of Medicine and Pharmacy “Gr. T. Popa”, 700115 Iasi, Romania; mindru.dana@umfiasi.ro (D.E.M.); dana.anton@umfiasi.ro (D.-T.A.-P.); ad.heidi91@gmail.com (H.A.); acluca@yahoo.com (A.C.L.); 2Department of Surgery II—Pediatric Surgery, University of Medicine and Pharmacy “Gr. T. Popa”, 700115 Iasi, Romania; 3Department of Genetics Endocrinology, National Institute of Endocrinology CI Parhon, 011863 Bucureşti, Romania; elenabraha@yahoo.com; 4Department of Biology, Faculty of Biology, “Alexandru Ioan Cuza” University of Iasi, 700505 Iasi, Romania; 5Department of Nursing, University of Medicine and Pharmacy “Gr. T. Popa”, 700115 Iasi, Romania; rosusolange@yahoo.com; 6Department of Surgery II, “Saint Spiridon” Hospital, 700115 Iasi, Romania; bernicvalik@yahoo.com; 7“Sfanta Maria” Emergency Children Hospital, 700309 Iasi, Romania; paduret.alexandra@gmail.com

**Keywords:** GLB1 mutation, GM1, aortopathies, fibroelastosis, lysosomal storage disease

## Abstract

Gangliosidosis (ORPHA: 79255) is an autosomal recessive lysosomal storage disease (LSD) with a variable phenotype and an incidence of 1:200000 live births. The underlying genotype is comprised GLB1 mutations that lead to β-galactosidase deficiency and subsequently to the accumulation of monosialotetrahexosylganglioside (GM1) in the brain and other organs. In total, two diseases have been linked to this gene mutation: Morquio type B and Gangliosidosis. The most frequent clinical manifestations include dysmorphic facial features, nervous and skeletal systems abnormalities, hepatosplenomegaly, and cardiomyopathies. The correct diagnosis of GM1 is a challenge due to the overlapping clinical manifestation between this disease and others, especially in infants. Therefore, in the current study we present the case of a 3-month-old male infant, admitted with signs and symptoms of respiratory distress alongside rapid progressive heart failure, with minimal neurologic and skeletal abnormalities, but with cardiovascular structural malformations. The atypical clinical presentation raised great difficulties for our diagnostic team. Unfortunately, the diagnostic of GM1 was made postmortem based on the DBS test and we were able to correlate the genotype with the unusual phenotypic findings.

## 1. Introduction

Lysosomal diseases (LSDs) are monogenic disorders with autosomal recessive or X-linked transmission that affect lysosomal function as a result of enzyme deficiencies, malfunctions of transporter proteins, or other proteins involved in lysosomal homeostasis [1]. LSDs caused by enzyme deficiencies are classified according to the affected substrate into glycogen storage diseases, mucopolysaccharidoses, sphingolipidoses, and glycoproteinoses [1].

Mainly, inborn errors of lysosomal metabolism cause severe neurological disorders, accompanied by visceromegaly and skeletal damage. However, any organ system can be affected, and the severity spectrum ranges from mild to severe. The manifestations of lysosomal diseases are due to cellular insults caused by the accumulated substrate, as well as autophagy anomalies, immune system dysfunctions, inflammatory processes, and oxidative stress [2]. The presentation of these anomalies depends on the substrate involved and the triggered cascades of intracellular and intercellular reactions. However, one enzyme may be involved in more than one substrate metabolism, leading to overlapping phenotypes, such as the *GLB1* gene encoding beta-galactosidase. In total, two diseases have been linked to *GLB1* mutations: Gangliosidosis type I (GM1) and Morquio type B (MBD). The former is caused by monosialotetrahexosylganglioside (GM1) accumulation in the brain and other organs, while the latter is a consequence of glycosamino-glycan (GAG) and keratan sulfate (KS) excess accumulation in bones, cartilage, and cornea [3]. Besides Morquio type B, gangliosidosis should be differentiated from mucopolysaccharidoses, other sphingolipidoses, and glycoproteinoses [4] (Table 1). 

Recent research indicates that Morquio type B and GMI are part of the spectrum of the same condition, and that the severity of the manifestations as well as overlapping cases is due to the localization of the *GLB1* gene mutation as well as the type of mutation [3]. The reported incidence for GM1 is 1:100,000–1:200,000 live births. Considering the rarity of the condition, and the recessive transmission, genetic advice is particularly important to prevent disease recurrence, especially in communities where a founder effect has been identified. 

## 2. Case Report

A 3-month-old infant was brought to the emergency room of our hospital with persistent respiratory infection with the onset in the first month of life and an evolution marked by short periods of lull, for which he received antibiotics and symptomatic treatment, to no avail. The personal history revealed difficult adaptation to extrauterine life that required non-invasive ventilation support and gavage feeding. There were also signs of fetal distress, and our patient was born with a low birth weight (2250 g) despite reaching full gestation time (39 weeks).

The clinical examination upon admission revealed altered general status, limb hypotonia, mild craniofacial dysmorphia, generalized edema, signs of respiratory distress (dyspnea, polypnea, intercostal retractions) with peripheral oxygen saturation of 85%, tachycardia with a heart rate of 180 bpm, grade II systolic murmur, and hepatosplenomegaly. Horizontal nystagmus was inconsistently noticed together with mild skeletal anomalies, such as conical fingers. 

Biological evaluations revealed hypochromic microcytic anemia, inflammatory syndrome (ESR = 17 mm/h, CRP = 20 mL/L), and hepatocytolisis (AST = 98 U/L, GGT = 118 U/L). Cardiac troponin T was 0.20 ng/mL and NT-proBNP levels remained constantly above 10,000 pg/mL. Blood cultures, urinalysis, and coprocytological examination did not reveal any pathologic changes. Considering the hepatosplenomegaly, we performed a bone marrow examination, which had normal results.

Thoracic radiography showed a retrocardiac triangular opacity, with medium intensity, a decrease in right pulmonary transparency with apical linear atelectasis, and minimal right costomarginal pleural reaction; the cardiothoracic index on the X-ray was 0.75.

Cardiothoracic CT revealed multiple foci of segmental pulmonary condensation, located at the level of both lungs and minimal bilateral pleural effusion (Figure 1).

The echocardiographic findings raised the suspicion of Bland-White-Garland syndrome (anomalous origin of the left coronary artery arising from the pulmonary artery), with dilated coronary arteries and signs of diastolic ventricular dysfunction with dilated cardiomyopathy. Minor valvular abnormalities were also identified—aortic bicuspidy and grade III mitral regurgitation—but there was no echocardiographic evidence of valvular thickening (Figure 2 and Figure 3).

In the presence of cardiac abnormalities and hepatosplenomegaly with minimal neurological signs, we then followed two diagnostic directions: hereditary cardiac pathology with cardiorespiratory decompensation and signs of acute respiratory failure, respectively, inborn error of metabolism.

In order to evaluate the presence of a metabolism error, readily available DBS tests were performed, which showed a beta-galactosidase value of 0.12 (normal values: 0.5–3.2 nmol/spot × 1 h), consistent of a significantly diminished beta-galactosidase activity. Testing for urinary levels of oligosaccharides and KS was not available at the time. 

A thoracic angio-CT was scheduled in order to evaluate the vascular anatomy of the coronary arteries and the possible other structural vascular abnormalities. However, our patient rapidly deteriorated, requiring orotracheal intubation and mechanical ventilation, as well as inotropic and hemodynamic support. The setting of MODS (multiple organ damage syndrome) with a pediatric SOFA score of three further complicated the prognostic of our patient, who eventually died. 

The macroscopic findings of the necropsy confirmed the presence of dilated cardiomyopathy, with a hypertrophic component at the level of the left ventricle and left heart fibroelastosis. Coronary arteries dilation and common trunk origin of the brachiocephalic and left common carotid artery from the aortic arch with hypoplasia of the carotid artery were also identified. Other macroscopic findings included hepatosplenomegaly, pulmonary congestion, and meningocerebral edema (Figure 4 and Figure 5). Thymus hypotrophy was also noted.

Microscopic evaluations highlighted the presence of foamy cells in the lungs, liver, spleen, and pancreas. Fibrosis lesions were described at the subepicardial levels in the left heart, as well as fibroelastosis markers. Inflammatory lesions with the presence of lymphocytes were identified in the cerebral, cardiac, pulmonary, and liver tissue samples (Figure 6, Figure 7, Figure 8 and Figure 9).

The medulogram was reexamined and a non-infiltrated hematogenous marrow, loaded with fat (adipocytes), with very rare pseudo-Niemmen-Pick cells was described. 

Based on the age of onset, cardiac phenotype, visceromegaly lack of significant skeletal anomalies, and the DBS tests results, our final diagnosis was infantile gangliosidosis.

## 3. Discussions

Gangliosidosis is a rare LSD with three subforms, depending on the age of onset and the severity of symptoms. It is mainly a neurodegenerative disease and typically infantile forms present with neurologic impairment as a result of brain structural damage and a lack of proper myelination. Late infantile forms manifest with structural and functional neurological damage with epileptic manifestations, EEG modifications, and atrophy lesions identified by a brain MRI [14]. Cases with an intermediate phenotype between Morquio B and Gangliosidosis have been described in the literature, with a borderline juvenile onset of symptoms, skeletal abnormalities, and abnormal urinary oligosaccharides levels [3]. 

Our patient matched the age criteria for infantile GM1, but the neurological abnormalities were not significant. Hypotonia was present but difficult to interpret as a disease sign considering the context of a respiratory distress syndrome accompanied by acute cardiac failure. 

The cardiac phenotype per se could have partially explained our patient’s symptoms even in the absence of a LSD, making diagnosis even more difficult. Primary fibroelastosis is known to cause respiratory failure and is associated with hypotonic neurologic disorders [15]. Studies have also found genetic mutations associated with pEFE, none of which are found anywhere near the *GLB1* gene. Our patient exhibited multiple congenital heart malformations—aortic bicuspid, dilated coronary artery, carotid artery hypoplasia, and common trunk origin of the brachiocephalic and left common carotid artery—supporting the growing body of literature regarding aortopathies associated with LSDs [16,17]. 

In our case, the significant visceromegaly was the key factor which eventually raised the suspicion of a lysosomal storage disease. To make matters more complicated, our patient had biological markers of liver dysfunction, which is usually correlated with MBD, but, considering the history of antibiotic administration [17], could not be interpreted as a disease sign [18,19]. 

Enzyme activity tests were conclusive for a *GLB1* related disease with infantile presentation and a grim prognostic. Unfortunately, the rapid demise of our patient did not allow for extensive genetic testing and molecular autopsy instruments are not readily available in our center, but we would strongly recommend them in such cases, whenever possible. 

It is worth mentioning the fact that our patient was of Roma ethnicity. Sinigerska et al. reported a high prevalence of R59H mutation among Roma ethnics in Bulgaria, incriminating a founder effect and suggesting that patients should be tested for GM1 if identified with symptoms consisting with LSDs [20]. 

There are limited treatment opportunities for GM1, and most of them are mainly supportive and symptomatic. Enzyme replacement therapy is available for other sphingolipidoses, but it has the disadvantage of not being able to cross the blood–brain barrier [21]. Substrate reduction therapy aims at inhibiting glycosphingolipid synthesis, but it acts in a non-discriminating fashion, affecting enzymes outside of the lysosomal homeostasis [19]. Gene therapy in GMI using adeno-associated vector virus (AAV) is becoming of interest due to its efficiency in transducing a wide range of cell types. Encapsulation of β-gal enzyme into artificial nanoparticles to traverse the blood–brain barrier has also been experimented in vitro [22,23,24,25]. In the infantile forms of GMI, life expectancy is nevertheless low and hampered by LQL because of multi-organ impairment. Our patient died within 21 days from admission, at the age of 4 months, making it impossible for any therapeutic measurements to be set in place, even if any were available. We would argue that neonatal screening of inborn error of metabolism should be expanded based on demographic, racial, and ethnic criteria, in order to prevent premature deaths and reach an acceptable level of a costs/benefit ratio [25]. In establishing screening programs, accessibility to medical services should be considered [26,27]. 

## 4. Conclusions

Inborn errors of metabolism are severe diseases with variable phenotype and unfavorable prognosis in the case of those with childhood onset. Cardiac manifestations, initially included in the category of cardiomyopathies and valvulopathies, tend to take on more complex forms, with a still unclear production mechanism and uncertain genetic substrate. Aortopathies associated with LSD are the subject of current research and the identification and establishment of an association pattern is particularly important because most of the time, at ages under 1 year, a severe cardiac anomaly can mask the presence of a lysosomal disease and delay the initiation of a specific therapy where it exists.

## Figures and Tables

**Figure 1 diagnostics-14-00491-f001:**
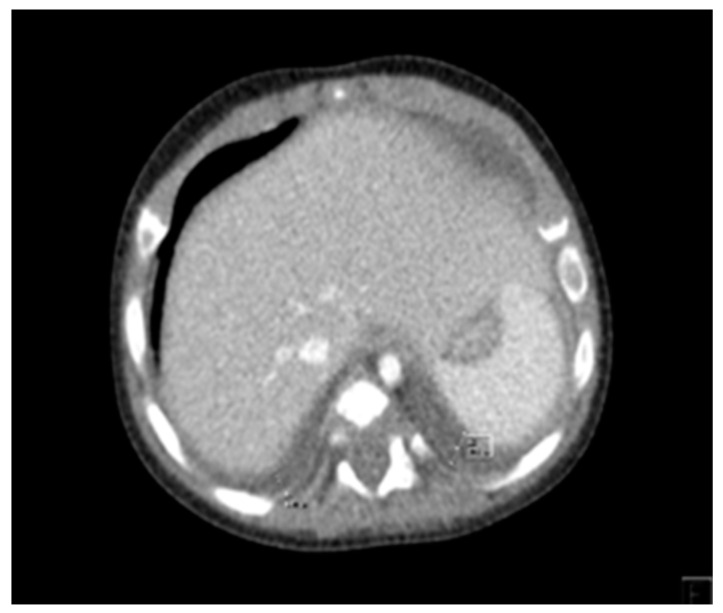
Thoracic CT showing minimal pleural effusion.

**Figure 2 diagnostics-14-00491-f002:**
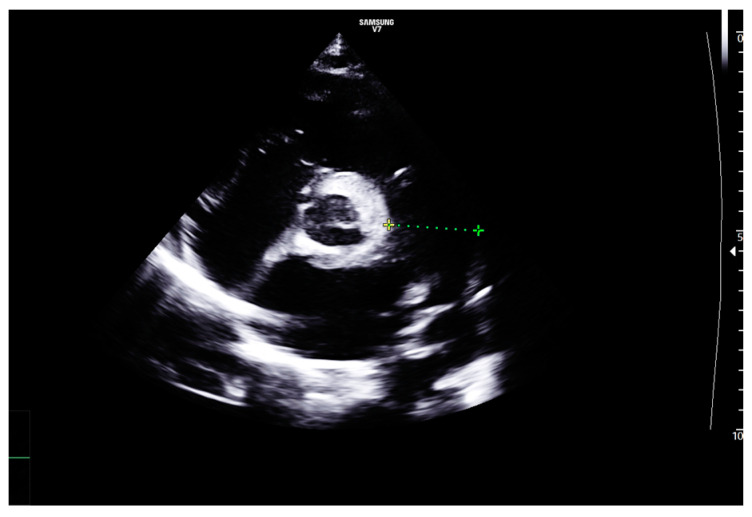
TTE. Parasternal short axis view. Aortic bicuspidy, dilated right coronary artery, dilated pulmonary trunk.

**Figure 3 diagnostics-14-00491-f003:**
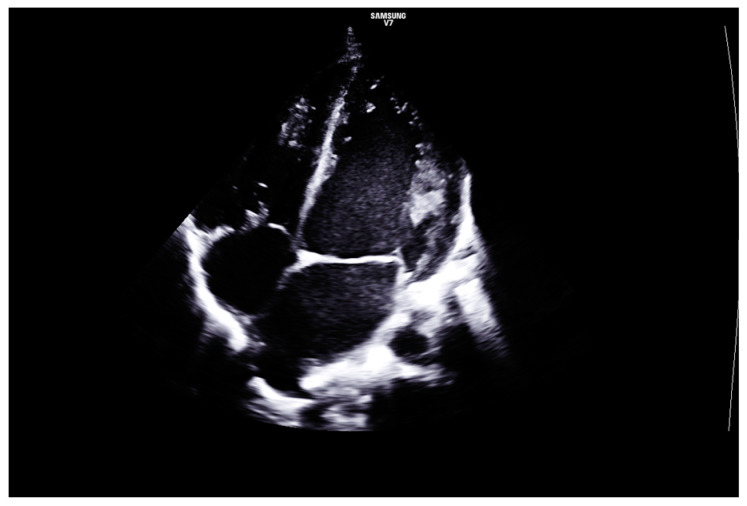
TTE. Apical 4 chambers view. Severely dilated left ventricle. No signs of valvular thickening.

**Figure 4 diagnostics-14-00491-f004:**
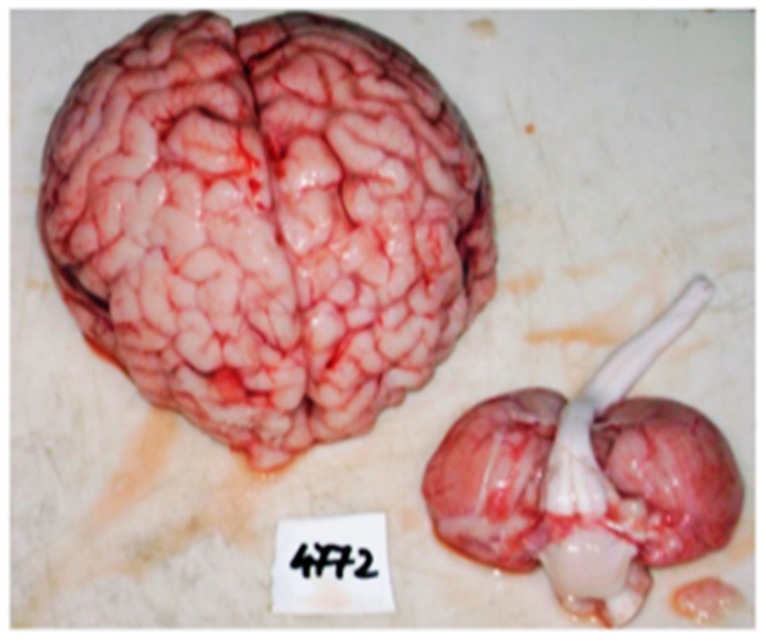
Meningocerebral edema.

**Figure 5 diagnostics-14-00491-f005:**
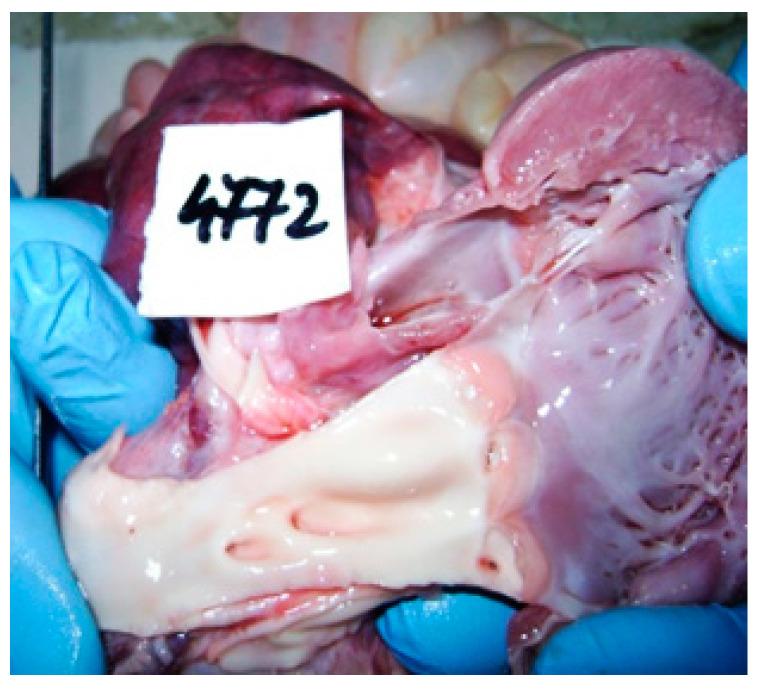
Dilated cardiomyopathy with LVH component and left heart fibroelastosis.

**Figure 6 diagnostics-14-00491-f006:**
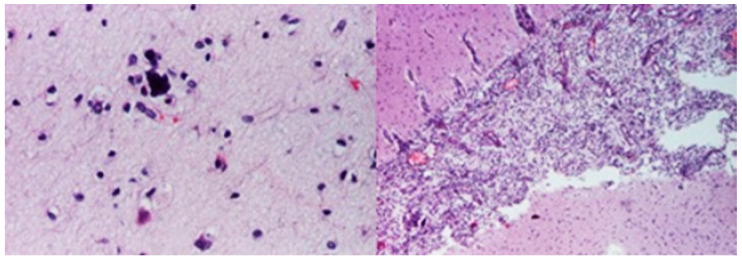
Meningitis (HE staining × 40)

**Figure 7 diagnostics-14-00491-f007:**
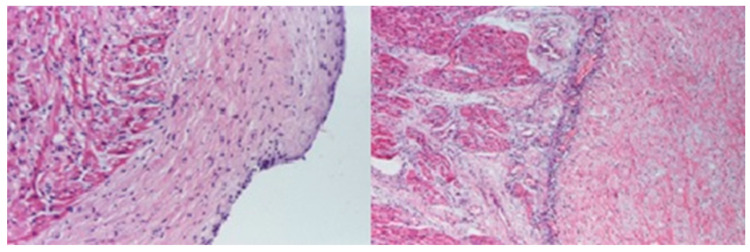
Myocardial fibroelastosis (HE staining × 40).

**Figure 8 diagnostics-14-00491-f008:**
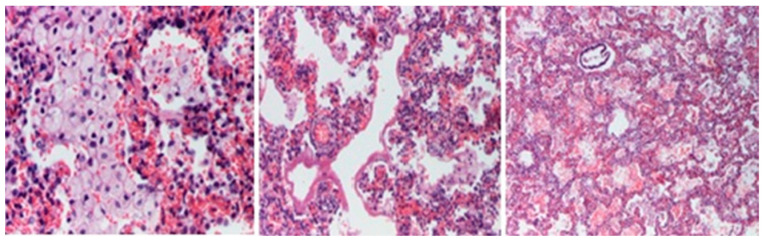
Endoalveolar foamy macrophages. Areas of hyaline membranes (HE staining × 100).

**Figure 9 diagnostics-14-00491-f009:**
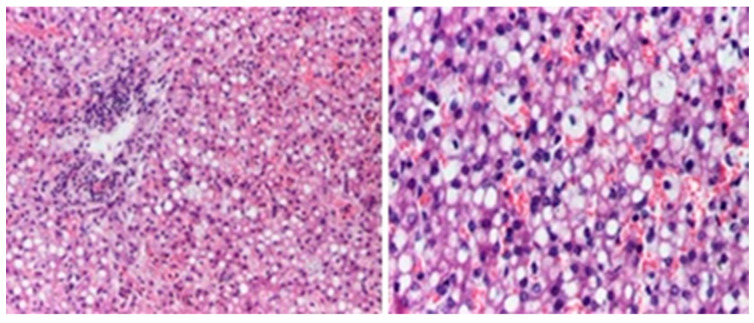
Liver with foamy cells, inflammatory cells in the porto-biliary spaces (HE staining × 200).

**Table 1 diagnostics-14-00491-t001:** Gangliosidosis differential diagnosis criteria with Morquio type B and other mucopolysaccharidosis, as well as glycoproteinosis. CC—corneal clouding; CRS—cheery red spots; EDD—early developmental delay; HCM—hypertrophic cardiomyopathy; DCM—dilated cardiomyopathy; KS—keratan sulfate; LC-MS—Liquid Chromatography-Mass Spectrometry; PC—pharmacological chaperones; ERT—enzyme replacing therapy [5,6,7,8,9,10,11,12,13].

	Gangliosidosis Type I			Morquio Type B (MBD)	Mucopolysaccharidosis (Except MBD)	Glycoproteinosis
Age of Onset	Infantile	Juvenile	10–30 Years	3–5 Years	Infantile/Juvenile	Infantile/ Juvenile/Adult
Facial features	Coarse	Mildly coarse	Dysmorphic or normal	-	Coarse	Coarse
Ocular	CRS	CC	CC	CC	CC; Glaucoma; Retinal dystrophy	CRS/CC
Skin	Mongolian spots	-	-	-	-	Angiokeratoma; Telangiectasia
Neurologic	EDD; Hypotonia Deafness; Blindness; Exaggerated startle response	Motor and speech impairment	Dystonia; Extrapiramidal signs	Normal	Hearing loss	Hearing loss; Ataxia; Seizures; Spasticity
Respiratory	Restrictive lung disease	-	-	Restrictive lung disease	Upper airway obstruction; Restrictive lung disease	-
Cardiovascular	HCM/DCM	HCM/DCM	HCM/DCM	Valvular defects	Valvular defects	Cardiomyopathies; Valvular anomalies
Digestive	Hepatosplenomegaly	Hepatosplenomegaly	-	Liver dysfunction	Hepatosplenomegaly; Diarrhea; Swallowing difficulties	Hepatosplenomegaly
Hematologic	Vacuolated lymphocytes	Vacuolated lymphocytes	Vacuolated lymphocytes	-	Anemia, thrombocytopenia, and coagulopathy in some diseases	Vacuolated lymphocytes; Immunodeficiency
Motor	+	+	+	-	-	+
Skeletal	Scoliosis; Dysostosis multiplex	Variable	Short stature	Dysostosis multiplex; Genu valgus; Short stature	Joint stiffness; Short stature	Dysostosis multiplex; Short stature
Renal	-	-	-	-	UTI and renal dysfunction possible	Renal failure
Cognitive	Severe impairment	Progressive decline	Intellectual disability	Normal intellect	Different stages of Intellectual disability	Progressive decline; Autism spectrum disorder
Non-genetic diagnostic tests	Urinary oligosaccharides; Elevated proinflammatory citokines; Enzyme activity	Urinary oligosaccharides; Elevated proinflammatory citokines; Enzyme activity	Urinary oligosaccharides; Elevated pro inflammatory citokines; Enzyme activity	Urinary excretion of KS by LC-MS; Elevated proinflammatory citokines; Enzyme activity	Urinary excretion of glycosaminoglycans by LC-MS; Enzyme activity	Urinary sialic acid-rich oligosaccharides; Enzyme activity
Genetic tests	GLB1	GLB1	GLB1	GLB1	Monogenic disorders	Monogenic disorders
Treatment options	Supportive and symptomatic; PC are under investigations	Supportive and symptomatic; PC are under investigations	Supportive and symptomatic; PC are under investigations	Supportive and symptomatic; PC are under investigations	ERT does not improve CNS symptoms; HSC transplant	Supportive and symptomatic; ERT available for alfa-Mannosidosis
Dietary strategies	Parenteral feeding/Enteral feeding using G-tubes		Ketogenic diet + Miglustat	Ketogenic diet + Miglustat	Antioxidants; Vitamin B6; Limited intake of sugar and milk	Oral zinc therapy in α-mannosidosis

## Data Availability

No new data were created or analyzed in this study. Data sharing is not applicable to this article.

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
