# Peer review of "Congenital Heart Malformations Masked by Infantile Gangliosidosis—Case Report and Growing Evidence for Metabolic Disease-Associated Aortopathies"

_diagnostics, 2024, doi:10.3390/diagnostics14050491_

Round 1

Reviewer 1 Report

Comments and Suggestions for Authors

This case is well written and certainly clinically interesting but there are some concerns which I hope the authors take into consideration.

General remark: what is the novelty of the present case? Dilated cardiomyopathy has been described earlier in gangliosidosis: Simma et al Klin Padiatr . 1990 May-Jun;202(3) and Lin et al. Acta Paediatr. 2000 Jul;89(7):880-3

Why was genetic testing not possible? If blood samples were taken DNA could still be extracted from residus.

Was ALCAPA ruled out??

Case report

Was CRP measured? NT pro BNP? Troponines?  what was the heart rate? Was blood pressure measured?

What was the cardio-thoracic index on Xray?

Was the origin of the left coronary normal on autopsy?

Author Response

Dear Reviewer,

Thank you very much for evaluating our manuscript. Your recommendations and comments have helped us improve our manuscript. Here we provide the requested corrections and address the comments. The changes we have made in the manuscript are highlighted in yellow.

  • General remark: what is the novelty of the present case? Dilated cardiomyopathy has been described earlier in gangliosidosis: Simma et al Klin Padiatr . 1990 May-Jun;202(3) and Lin et al. Acta Paediatr. 2000 Jul;89(7):880-3.
  • Response: We presented this case especially due to the rarity of the underlying metabolic disease, the incidence being 1/200,000 live newborns. Therefore, in the current study we present the case of a 3 months old male infant, admitted with signs and symptoms of respiratory distress alongside rapid progressive heart failure, with minimal neurologic and skeletal abnormalities, but with cardiovascular structural malformations. The atypical clinical presentation raised great difficulties for our diagnostic team. 
  • Why was genetic testing not possible? If blood samples were taken DNA could still be extracted from residus.
  • Response:  Although the parents gave their consent for investigations, treatment, necropsy and for the presentation of the case in scientific journals, they did not want the continuation of post-mortem genetic investigations.
  • Was ALCAPA ruled out??
  • Response: Yes, ALCAPA was ruled out at necropsy exam.
  • Was CRP measured? NT pro BNP? Troponines?  what was the heart rate? Was blood pressure measured? What was the cardio-thoracic index on Xray?
  • Response: We added in the manuscript the requested information.
  • Was the origin of the left coronary normal on autopsy?
  • Response: Coronary arteries were dilated but the origin was normal.

Thank you again for reviewing our manuscript. 

Reviewer 2 Report

Comments and Suggestions for Authors

Mindru et al present a case report of a 3-month-old  infant with cardiovascular manifestation of gangliosidosis.

1. Could you please add a picture from the echocardiographic exam? I think it would be of high interest to the readers.

2. I think the focus of the case is that the gangliosidosis had an atypical clinical presentation affecting the cardiovascular system. Therefore, more figures illustrating the CV system would be necessary in order to increase the knowledge of this possible clinical presentation and help the readers recognize it. I think it would be a good idea to add images from the CT angio.

Author Response

Dear Reviewer,

Thank you very much for evaluating our manuscript. Your recommendations and comments have helped us improve our manuscript. Here we provide the requested corrections and address the comments. The changes we have made in the manuscript are highlighted in yellow.

  • Could you please add a picture from the echocardiographic exam? I think it would be of high interest to the readers.
  • Response: Figure 2 and 3 are from echocardiographic exam, showing Aortic bicuspidy, dilated right coronary artery, dilated pulmonary trunk (TTE. Parasternal short axis view) and Severely dilated left ventricle, No signs of valvular thickening (TTE. Apical 4 chambers view).
  • I think the focus of the case is that the gangliosidosis had an atypical clinical presentation affecting the cardiovascular system. Therefore, more figures illustrating the CV system would be necessary in order to increase the knowledge of this possible clinical presentation and help the readers recognize it. I think it would be a good idea to add images from the CT angio.
  • Response: Thoracic angio-CT was scheduled in order to evaluate the vascular anatomy of the coronary arteries and the possible other structural vascular abnormalities. However, our patient rapidly deteriorated, requiring orotracheal intubation and mechanical ventilation, as well as inotropic and hemodynamic support. The setting of MODS (multiple organ damage syndrome) with a pediatric SOFA score of 3 further complicated the prognostic of our patient, who eventually died.

Round 2

Reviewer 1 Report

Comments and Suggestions for Authors

All questions were answered

Reviewer 2 Report

Comments and Suggestions for Authors

The authors have adequately addressed my comments and the manuscript has improved.